# Accurate Long-Read RNA Sequencing Analysis Reveals the Key Pathways and Candidate Genes under Drought Stress in the Seed Germination Stage in Faba Bean

**DOI:** 10.3390/ijms25168875

**Published:** 2024-08-15

**Authors:** Xin Wen, Changyan Liu, Fangwen Yang, Zhengxin Wei, Li Li, Hongwei Chen, Xuesong Han, Chunhai Jiao, Aihua Sha

**Affiliations:** 1MARA Key Laboratory of Sustainable Crop Production in the Middle Reaches of the Yangtze River (Co-Construction by Ministry and Province)/Engineering Research Center of Ecology and Agricultural Use of Wetland of Ministry of Education, College of Agriculture, Yangtze University, Jingzhou 434025, China; wenxinn0205@163.com (X.W.); hcwzx999@163.com (Z.W.); 2Institute of Food Crops, Hubei Academy of Agricultural Sciences/Hubei Key Laboratory of Food Crop Germplasm and Genetic, Wuhan 430064, China; liucy0602@hbaas.com (C.L.); hslili07@163.com (L.L.); hwchen25@126.com (H.C.); hanxuesong@hbaas.com (X.H.); 3Shanghai Agrobiological Gene Center, Shanghai 201106, China; fangwen_yang@163.com

**Keywords:** faba bean, drought response, PacBio and Illumina RNA sequencing, differentially expressed genes, aspartate aminotransferase

## Abstract

Faba bean is an important pulse. It provides proteins for the human diet and is used in industrial foodstuffs, such as flours. Drought stress severely reduces the yield of faba bean, and this can be efficiently overcome through the identification and application of key genes in response to drought. In this study, PacBio and Illumina RNA sequencing techniques were used to identify the key pathways and candidate genes involved in drought stress response. During seed germination, a total of 17,927 full-length transcripts and 12,760 protein-coding genes were obtained. There were 1676 and 811 differentially expressed genes (DEGs) between the varieties E1 and C105 at 16 h and 64 h under drought stress, respectively. Six and nine KEGG pathways were significantly enriched at 16 h and 64 h under drought stress, which produced 40 and 184 nodes through protein–protein interaction (PPI) analysis, respectively. The DEGs of the PPI nodes were involved in the ABA (abscisic acid) and MAPK (mitogen-activated protein kinase) pathways, N-glycosylation, sulfur metabolism, and sugar metabolism. Furthermore, the ectopic overexpression of a key gene, *AAT*, encoding aspartate aminotransferase (AAT), in tobacco, enhanced drought tolerance. The activities of AAT and peroxidase (POD), the contents of cysteine and isoleucine, were increased, and the contents of malonaldehyde (MDA) and water loss decreased in the overexpressed plants. This study provides a novel insight into genetic response to drought stress and some candidate genes for drought tolerance genetic improvements in this plant.

## 1. Introduction

Faba bean is an annual dicotyledonous pulse that is widely grown in Asia, Africa, and the Mediterranean region [1]. The current production of faba bean worldwide is 5.7 million tonnes, which is modest compared to soybean (353 million tonnes) and pea (14.6 million tonnes) [2]. Faba bean contains high protein levels, at approximately 250 g protein/kg seed, which is higher than most pulses, such as peas, chickpeas, lentils, and beans [3]. Therefore, protein from faba bean has been applied to improve the functional properties of ingredients such as flours, concentrates, and isolates as industrial materials [4]. A complement of faba bean flour in pasta can improve the protein quality, bioavailability, baking quality, and starch digestibility due to the increased protein concentration [5]. In addition, faba bean seeds are rich in health-promoting constituents, such as phenolic compounds, resistant starch, dietary fibers, and non-protein amino acids (L-DOPA, GABA), which offer the potential to develop new nutraceuticals and biofunctional food ingredients [6]. Further, faba bean can be used as a rotational or intercropping crop to increase the yield of barley and wheat due to its high ability to fix atmospheric nitrogen [6]. However, the yield of faba bean is not as stable as many other legumes [7]. On the other hand, genetic improvements in faba bean face great challenges due to its mixed breeding system and giant genome size (~13 Gb) with over 85% repetitive DNA [8]. Recently, a high-quality chromosome-scale assembly of the faba bean genome was conducted [9], and several transcriptomic studies were also performed [10,11,12,13], which accelerate the speed of faba bean genomics-based breeding. NGS (next-generation sequencing) technique shows great efficiency and speed in gene discovery. However, it cannot accurately predict each isoform due to the produced short sequencing reads [14,15,16]. PacBio SMRT (single-molecule, real-time sequencing technology) can yield kilobase-sized sequence reads for full-length mRNA molecules, but with a higher rate of errors [17,18,19]. Thus, combining the SMRT and NGS techniques can generate a more complete full-length transcriptome with less errors. Therefore, in this study, these new sequencing techniques will be applied to perform the full-length transcriptome sequencing of faba bean under drought stress at the seed germination stage.

Drought stress is one of the most serious abiotic stresses, greatly impacting crop production and frequently occurring due to global climate change [20]. A high rate and uniformity of seed germination are two critical factors for good seedling establishment, growth, and productivity in a crop under drought stress [21]. Unfortunately, seed germination is first and foremost impacted by drought stress [20]. Faba bean requires an abundant amount of water to grow and develop, especially during seed germination. Severe drought stress can cause non-uniform germination or non-germinating seeds after sowing, thus leading to a serious production loss or even no harvest. Until now, there have only been a few reports on the genetic basis of drought stress tolerance at seed germination. For instance, 338 single-nucleotide polymorphisms (SNPs) were detected under drought associated with seed germination-related traits in barley [21]. In total, 39 quantitative trait loci (QTLs) were identified as being related to germination under drought stress in *Brassica napus* [22]. The ectopic overexpression of a sheepgrass transcription factor, *LcMYB2*, could promote seed germination under drought stress in Arabidopsis [23]. Notably, the phenylpropanoid-related pathway regulated drought resistance during germination in foxtail millet [24]. However, the genetic mechanism of the faba bean response to drought stress during the seed germination is still not clear. Therefore, it is essential to identify the key pathways and candidate genes involved in drought stress at seed germination in faba bean.

In this study, to unravel the new pathway and candidate genes involved in the drought stress response to drought stress during seed germination, combined PacBio and NGS techniques were applied to perform the full-length transcriptomic analyses of two faba bean varieties, E1, a drought-sensitive Chinese local variety, and C105, a drought-stress-tolerant variety from Ethiopia, under drought stress at seed germination. These findings offer a novel insight into the key pathways and candidate genes for drought stress genetic improvement during seed germination in this species.

## 2. Results

### 2.1. The Phenotypes of E1 and C105 under Drought Stress at the Germination Stage

To determine the appropriate time points at which the key genes regulate seed germination, the germinating process of E1 and C105 was investigated under control and drought stress conditions. The germination rate of both E1 and C105 was approximately 50% under control condition at 48 h. C105 began to germinate at 64 h, whereas E1 did not germinate until 96 h under drought stress (Figure 1). Further, the germination rate of C105 reached 45.2%, whereas that of E1 was 24.3% at 168 h under drought stress, suggesting that 64 h of drought stress at seed germination could distinguish the drought-tolerant variety C105 from the drought-sensitive variety E1. Hence, the 16 h and 64 h seed germination points for E1 and C105 were selected to investigate their genome-wide gene expression profiles under control and drought stress conditions.

### 2.2. Identification of Full-Length Transcripts and Annotation under Drought Stress

A total of 351,263 CCS (circular consensus sequence) reads were identified from 7,530,616 subreads (Appendix A). CCS reads contained 119,989 full-length reads with an average length of 2151 bp. A total of 313,565 high-quality consensus isoforms and low-quality consensus isoforms were corrected with Illumina sequencing data (Appendix A). Finally, 65,346 corrected full-length isoforms were used for further analysis after removing the redundant isoforms. BUSCO analysis showed that the percentage of complete genes is 57.2%, with 25.3% complete single-copy genes and 31.9% complete duplicated sequence (Appendix A). The mean length of the full-length transcripts was 1019 bp, with a range of 200 to 7023 bp. In total, 17,927 unigenes, including 12,760 protein-encoding genes, were detected from the full-length transcripts. By blasting against the public reference genome for faba bean [9], 15,037 out of 17,927 unigenes could be hit, whereas 2890 unigenes did not match. In addition, 2371 out of the 15,037 unigenes that were hit could be multiplied to match with the reference genome (Appendix A).

The protein-encoding genes were annotated with the public databases, and 7946 (62.23%) were annotated to the five databases (Appendix A). Meanwhile, 12,509 out of the 12,760 unigenes were assigned to 22 functional categories by KOG annotation (Appendix A), which had hits with GO terms (Appendix A). KEGG annotation showed that the most enriched transcripts were in the categories of metabolic pathway (3310), genetic information processing (1527), organismal systems (1419), cellular component (1000), and environmental information processing (749), respectively (Appendix A).

### 2.3. Identification of Differentially Expressed Genes (DEGs) of C105 and E1 under Drought Stress

To assess the gene expression profiles of faba bean under drought stress, the Illumina reads were collected to determine gene expression levels. A total of 1961.44 million clean reads were obtained for the Illumina libraries (Appendix A). For the comparison of the different responses of E1 and C105 to drought stress, a total of 1676 DEGs (1006 and 670 up-/down-regulated) and 811 DEGs (463 and 348 up-/down-regulated) were identified for the comparison of T1_16 vs. T2_16 and T1_64 vs. T2_64, respectively. Likewise, 570 and 560 DEGs were detected under normal condition in the comparison of CK1_16 vs. CK2_16 and CK1_64 vs. CK2_64, respectively (Figure 2A, Appendix A). Overall, the number of DEGs was much greater in germinated seeds under drought stress than that under normal conditions between E1 and C105, indicating that more DEGs were induced by drought stress at the seed germination stage. Furthermore, more DEGs were detected in germinating seeds at 16 h than that at 64 h under drought stress, whereas this amount was nearly equal under normal conditions, suggesting that more DEGs were induced at the early stage of seed germination. Importantly, 167 DEGs were detected in both T1_16 vs. T2_16 and in CK1_16 vs. CK2_16. Meanwhile, 159 DEGs were detected in both T1_64 vs. T2_64 and in CK1_64 vs. CK2_64 (Figure 2B). In total, 225 DEGs were detected both in T1_16 vs. T2_16 and in T1_64 vs. T2_64, implying that these DEGs play crucial roles in regulating drought stress response from the early-to-late germination stages.

Next, we compared the DEGs of E1 and C105 in different response to drought stress at different time points in seed germination. In E1, 888 and 546 DEGs were up-/down-regulated in the comparison of T1_16 vs. T1_64, whereas there were 892 and 476 up-/down-regulated genes in CK1_16 vs. CK1_64 (Figure 2A), respectively. In C105, the up-/down-regulated DEGs were 594 and 376 in the comparison of T1_16 vs. T1_64, but there were 2840 and 1801 in the comparison of CK2_16 vs. CK2_64, respectively. The number of DEGs (4641) in C105 was much higher than that in E1 (1368) in the comparison of 16 h and 64 h under control conditions. However, the number of DEGs was lower in C105 than that in E1 (970/1434) under drought stress. We inferred that the seed germinating processes was initiated earlier in E1 compared to C105 under control conditions, in which there were more expressed genes involved in the regulation of seed germination earlier in E1, which led to fewer DEGs in the comparison of 16 h to 64 h. Conversely, the seed germinating processes initiated later in E1 than C105 under drought stress, which resulted in the production of more DEGs related to the regulation of seed germination progress. This finding was consistent with the statistics for the germination rate, in which the germination rate of E1 was higher than that of C105 at 64 h in CK whereas it was lower under drought stress (Figure 1B). A total of 1069 DEGs were detected in both CK1_16 vs. CK1_64 and in CK2_16 vs. CK2_64 (Figure 2C), and these might be involved in the fundamental time-course regulation of seed germination progress. Meanwhile, 191 DEGs were detected both in T1_16 vs. T1_64 and in T2_16 vs. T2_64 (Figure 2C); these DEGs probably regulated the drought stress response at seed germination.

### 2.4. Pathway Enrichment Analysis of DEGs at 16 h and 64 h of Drought Stress

To continue, six pathways were significantly enriched with the DEGs of T1_16 vs. T2_16, including cysteine and methionine metabolism (ko00270), N-glycan biosynthesis (ko00510, ko00513), sulfur metabolism (ko00920), ABC transporters (ko02010), and the MAPK pathway (ko04016) (Appendix A, Appendix A). In total, 22 and 18 DEGs were involved in cysteine and methionine metabolism and the MAPK pathway, respectively. In particular, for the ABA signaling and MAPK pathway, five DEGs (DN27766_c0_g1, DN28548_c0_g1, DN34399_c0_g1, DN21659_c0_g1, and DN24381_c0_g1) encoded the ABA (abscisic acid) receptor PYL (Pyrabactin-resistance like), one (DN8484_c0_g1) encoded the ABA signaling protein phosphatase 2c (PP2C), and one (DN12095_c0_g1) encoded SnRK2 (SNF1_SCHPO SNF1-like protein kinase ssp2).

Nine KEGG pathways were significantly enriched with the DEGs of T1_64 vs. T2_64 (*p* < 0.05), including the biosynthesis of secondary metabolites (ko01110), the pentose phosphate pathway (ko00030), metabolic pathways (ko01100), fructose and mannose metabolism (ko00051), galactose metabolism (ko00052), taurine and hypotaurine metabolism (ko00430), cutin, suberine and wax biosynthesis (ko00073), the biosynthesis of amino acids (ko01230), and SNARE interactions in vesicular transport (ko04130) (Appendix A, Appendix A). The metabolic pathways and biosynthesis of secondary metabolites were prevalent, with 164 and 100 DEGs enriched, respectively. Several DEGs were reported to participate in the regulation of drought stress, such as DN4242_c0_g1 (encoding beta-amylase) in the biosynthesis of secondary metabolites [25], DN5379_c0_g1 (encoding glucose-6-phosphate 1-dehydrogenase) in the pentose phosphate pathway [26], and DN1127_c0_g1 (encoding raffinose synthase) in the galactose metabolism pathway [27].

Meanwhile, 16 and 19 pathways were significant enriched with *p* value < 0.05 in the comparison of T1_16 vs. T1_64 and T2_16 vs. T2_64, respectively (Appendix A). Metabolic pathways (ko01100) and the biosynthesis of secondary metabolites (ko01110) were the main pathways for both T1_16 vs. T1_64 and T2_16 vs. T2_64, inferring that these genes involved in the two pathways regulated seed germination in both drought-tolerant and drought-sensitive varieties under drought stress. GO enrichment analyses were performed with all the DEGs. In terms of biological processes, the DEGs in all comparisons under drought stress were enriched in several abiotic response processes, such as the responses to temperature, cold, cytokinin, oxygen-containing compounds, inorganic substances, chemicals, alcohol, and abscisic acid. In terms of molecular function, protein kinase activity, signal transduction, and receptor activity were mainly present in T1_16 vs. T2_16, whereas the main molecular functions were oxidoreductase activity, structural molecule activity, the structural constituent of ribosomes, and isomerase activity in T1_64 vs. T2_64. Moreover, catalytic activity and transporter activity were the main items both in T1_16 vs. T1_64 and in T2_16 vs. T2_64 (Appendix A).

### 2.5. PPI Network Analysis of DEGs at 16 h and 64 h of Drought Stress

To clarify the genetic regulation of drought stress in faba bean, we investigated the PPI network of the DEGs detected in T1_16 vs. T2_16 and in T1_64 vs. T2_64, which accounted for the different drought responses between E1 and C105. Accordingly, the PPI networks of DEGs in T1_16 vs. T2_16 and in T1_64 vs. T2_64 were constructed. The 40 nodes with the DEGs in T1_16 vs. T2_16 generated four interaction sub-networks, which were categorized into the following four types of pathways: N-glycan biosynthesis, cysteine and methionine metabolism, sulfur metabolism, and the MAPK pathway (Figure 3A, Appendix A). Based on the above results for PPI in DEGs-encoding proteins, we found that the DEGs functioned in the regulation of the ABA and MAPK pathways, N-glycosylation, TCA cycle, the biosynthesis and metabolism of H_2_S and sulfated compounds, methionine and isoleucine synthesis, ethylene synthesis, etc. Under drought stress, the ABA and MPKK pathways induced the differential gene expressions of methionine, ethylene, and sulfated compound biosynthesis, indicating that kinase and plant hormone signaling (ABA, ethylene) affected amino acid and carbohydrate metabolism under drought stress (Figure 3B). 

Meanwhile, a total of 184 nodes in T1_64 vs. T2_64 were clustered into two interaction sub-networks, that is, metabolic pathways and ethylmaleimide-sensitive factor adaptor protein receptor (SNARE) interactions in vesicular transport (Appendix A, Appendix A). The hub genes that interacted with SNARE proteins and the hub-interacting genes involved in the pentose phosphate pathway, fructose and mannose metabolism, galactose metabolism, and the biosynthesis of amino acids were simplified, to be presented in Figure 4A, by removing other proteins, which included 56 hub genes. The 56 hub genes participated in the regulation of TCA cycles, the pentose phosphate pathway (PPP), the Suc catabolism pathway, the glycolysis pathway, sulfur metabolism, SNAREs, the urea cycle, etc. These hub genes in the PPI network showed the constant regulation of carbohydrate metabolism in the late stage as well as in the early stage of drought stress, highlighting similar regulation of the drought stress response process. Nevertheless, they also show the specific regulation of the urea cycle, fatty acid oxidation, and IAA biosynthesis, suggesting a specific drought tolerance response (Figure 4B,C). Therefore, 56 hub genes in the PPI network could be candidates for further functional studies.

To confirm the transcriptomic results, we selected 17 and 20 hub genes in PPI network analysis from the comparisons of T1_16 vs. T2_16 and T1_64 vs. T2_64 to determine their expression levels by qRT-PCR, respectively. The expression levels of most selected genes were consistent between RNA-seq data and qPCR detection (Appendix A) with the exception of DN8484_c0_g1, DN19743_c0_g1, and DN25373_c0_g1 in the comparison of T1_16 vs. T2_16, and DN21391_c0_g1, DN29210_c0_g1, DN13680_c0_g1, and DN7026_c0_g1 in the comparison of T1_64 vs. T2_64, suggesting that the RNA-seq data were relatively reliable in this study.

### 2.6. Ectopic Overexpression of AAT Enhanced Drought Tolerance in Tobacco

At present, there is lack of efficient transformation methods in faba bean (Björnsdotter et al., 2021) [13]; we adopted the PVX (*Potato mosaic virus X*) vector to ectopically overexpress a faba bean gene in *N. benthamiana* to verify the function of candidate genes under abiotic stress. As a routine method, target protein can be highly accumulated by overexpressing a foreign gene via the PVX vector and moving away from initially infected cells to uninfected ones. Therefore, the PVX-based vector can be used to identify the function of the candidate genes involved in the abiotic response [28]. 

Therefore, in this study, one DEG (DN8484_c0_g1) gene, encoding aspartate (Asp) aminotransferase (AAT), was up-regulated at both 16 h and 64 h in C105 under drought stress, which was linked the MAPK pathway and sugar metabolism with cysteine and methionine metabolism in the PPI network (Figure 3A and Figure 4A). Asp is a precursor for the biosynthesis of nucleotides, nicotinamide adenine dinucleotide, organic acids, amino acids, and a constitution of protein required for plant growth and defense [29]. AAT catalyzes a reversible transamination between glutamate and oxaloacetate to yield Asp and 2-oxoglutarate. Here, the selected *AAT* gene was ectopically overexpressed by means of the PVX vector in *N. benthemiana*, and drought tolerance was then evaluated. 

The results show that the ectopic overexpression of *AAT* obviously enhanced the drought tolerance of transgenic tobacco plants. The newly developed leaves grew normally in plants ectopically overexpressing *AAT*, but they wilted in the empty vector-inoculated controls after 11 d under drought stress (Figure 5A). The expression level of the *AAT* gene was detected only in *AAT*-overexpressed plants by RT-PCR (Figure 5B), implying that overexpressed *AAT* enhanced drought tolerance in tobacco. The relative water loss was significantly lower in *AAT*-overexpressed plants than that were inoculated with the empty vector and control under drought stress. However, there was no difference between the empty-vector and *AAT-*overexpressed plants before drought stress (Figure 5C,D). 

Furthermore, the enzymatic activity of AAT in tobacco plants with ectopic overexpression of the *AAT* gene was increased compared to the uninoculated plants or empty-vector-inoculated plants under both 0 d and 10 d post drought stress, which was more greatly enhanced by drought stress (Figure 5E). The content of malonaldehyde (MDA) and the activity of peroxidase (POD) were reduced and increased, respectively, in the leaves of tobacco plants with overexpressed *AAT* under drought stress (Figure 5F,G). However, there was no change in the activity of SOD (Figure 5H). The contents of aspartate and proline in *AAT*-overexpressed tobacco plants were significantly enhanced compared to control or empty-vector-inoculated plants before drought stress, whereas there were no significant differences among them under drought stress (Figure 5I,N; Appendix A). The content of arginine was significantly higher or lower in *AAT*-overexpressed plants than that in control or empty-vector-inoculated plants before or after drought stress treatment, respectively (Figure 5L, Appendix A). The contents of glutamate and glycine showed no difference among the control, empty vector, and *AAT*-overexpressed tobacco plants, but it was significantly higher or lower in *AAT-*overexpressed tobacco plants than that in control or empty-vector-inoculated plants under drought stress (Figure 5J,K; Appendix A). The content of cysteine and isoleucine was significantly higher in *AAT*-overexpressed tobacco plants than that in the control or empty-vector-inoculated plants under drought stress (Figure 5M,O; Appendix A). 

These above results showed that the ectopic overexpression of the *AAT* gene in tobacco enhanced the drought stress by increasing the activities of AAT and peroxidase (POD) and the contents of cysteine and isoleucine and reducing the content of MDA and water loss. Therefore, *AAT* could be identified as one candidate gene to potentially be applied in drought tolerance genetic improvements in faba bean.

## 3. Discussion

In this study, the accurate long-read RNA sequencing analysis indicated that a number of DEGs were involved in the response to drought stress in the germinating seeds of faba bean. The DEGs were mainly related to cysteine, methionine, and sulfur metabolisms and the MAPK pathway at the early stage of seed germination (16 h), and metabolic pathways and the biosynthesis of secondary metabolites at the late stage of germination (64 h), based on KEGG pathway enrichment analysis (Appendix A). PPI analysis demonstrated that 40 and 56 hub genes played essential roles in the regulation of the seed germination response to drought stress (Figure 3 and Figure 4). Notably, some DEGs had been reported to regulate drought stress in other species, such as D-cysteine desulfhydrase 2 (DN11736_c0_g1), adenylylsulfate reductase (DN1076_c0_g1), aldehyde dehydrogenase (DN5655_c0_g1, DN11266_c0_g1), raffinose synthase (DN1127_c0_g1), tryptophan synthase (DN7405_c0_g1), and cystathionine beta-synthase (DN13237_c0_g1, DN11468_c0_g1) [30,31,32,33,34]. Meanwhile, some genes were responsive to other abiotic stress, such as aspartate aminotransferase (DN8046_c0_g1, DN9977_c0_g1), malate dehydrogenase (DN3638_c0_g1, DN14639_c0_g1), transaldolase (DN15076_c0_g1), triosephosphate isomerase (DN15197_c0_g1), and fructokinase (DN31004_c0_g1) [35,36,37,38,39]. These genes might play conserved roles in regulating drought and other abiotic stresses in plants.

Furthermore, KEGG pathway enrichment and PPI network analyses were performed in this study. Here, seven genes were found to be involved in the ABA signaling and MAPK pathway, including the ABA receptor (DN34399_c0_g1, DN27766_c0_g1, DN21659_c0_g1, DN24381_c0_g1), phosphatase 2c (DN8484_c0_g1), SNF1-like protein kinase 2 (DN12095_c0_g1), and mitogen-activated protein kinase (DN25802_c0_g1), respectively. ABA is a vital phytohormone that suppresses seed germination by antagonistically and synergistically interacting with GA hormone [40]. Moreover, PYL, PP2C, and SnRK2 are essential protein partners in regulating ABA perception and signal relay [41]. The MAPK cascade also co-operates with the ABA signaling pathway during seed germination [42].

Significantly, a large number of DEGs were newly identified as being responsive to drought stress at seed germination. These new DEGs were identified as being involved in the metabolisms of Asp, sugars, protein N-glycosylation, and the membrane trafficking system. For example, Asp is a key metabolic hub interconnecting the metabolic pathways of Asp family amino acids, proteins, the TCA cycle and glycolysis pathway, and hormonal conjugates [29]. Two unigenes, encoding AAT (DN8046_c0_g1, DN9977_c0_g1), were up-regulated in the germinating seed at 16 h or 64 h. Carbohydrates serve as storage molecules for germination in seeds, of which raffinose family oligosaccharides (RFOs) and sucrose (*Suc*) are two key constitutes [43]. The sucrose and starch were metabolized to ATP and low-molecular-weight molecules required for biosynthesis and nitrogen assimilation via the glycolysis pathway and TCA cycle [44,45]. A number of DEGs were detected as being involved in sugar metabolism, such as DN1127_c0_g1, DN19804_c0_g1, DN4147_c0_g1, and DN8762_c0_g1 associated with RFO metabolism, DN6027_c0_g1, DN11132_c0_g1, DN15560_c0_g1, DN10457_c0_g1, DN12702_c0_g1, and DN5157_c0_g1 related to the glycolysis pathway, DN3638_c0_g1, DN14639_c0_g1, DN16771_c0_g1, DN6507_c0_g1, DN20413_c0_g1, DN6030_c0_g1, and DN244_c0_g1 related to the TCA cycle, and DN5379_c0_g1, DN4147_c0_g1, DN13680_c0_g1, DN15076_c0_g1, and DN8762_c0_g1 in relation to the PPP [46]. Protein N-glycosylation involves the dolichol-linked oligosaccharide (DLO) precursor synthesis, the final oligosaccharide transferring, and the protein N-glycans modification steps [47,48]. Eight DEGs (DN25373_c0_g1, DN9307_c0_g1, DN11712_c0_g1, DN15532_c0_g1, DN7882_c0_g1, DN1381_c0_g1, DN1271_c0_g1, and DN1742_c0_g1) were identified in seeds at the early germination stage. The membrane trafficking system SNARE mediates the fusion between vesicles and target membranes, which show multiple roles in biotic and abiotic stress tolerance and cellular stimulus response in plants [49]. In this study, seven *SNAREs* were up-/down-regulated in the seeds at the late stage of germination (64 h), suggesting that protein biosynthesis and transport participate in the seed germination process.

Based on the above findings, a schematic model was presented to explain the genetic regulation mechanism of faba bean response to drought stress (Figure 6). At the early stage of seed germination (16 h after drought stress), drought stress induces the ABA signaling pathway, which probably initiates MAPK cascades and sequentially phosphorylates aspartate biosynthesis in germinating seeds. The aspartate then serves as a precursor to synthesize methionine, cysteine, and oxaloacetate, which are further involved in the metabolisms of ethylene, biotin, polyamines, H_2_S, the tricarboxylic acid (TCA) cycle, etc. In addition, sulfur metabolism is induced to produce sulfated compounds and H_2_S, and then H_2_S regulates the biosynthesis of polyamines and sugars. The sugars are mainly metabolized by glycolysis and PPPs to produce some other metabolites, such as histone, nucleotides, IAA, and RFOs, to cope with drought stress at the later stage of germination (64 h after drought stress at seed germination). Meanwhile, the oxidation of fatty acids, the TCA cycle, and the photorespiration pathway are reduced to save energy, whereas the urea cycle, the biosynthesis metabolism of lysine, and aspartate re-cycles are enhanced to produce more amino acids. Protein N-glycosylation and SNAREs regulate the metabolisms of methionine and cysteine and the TCA pathway at the early and late stage of germination, respectively. A previous study showed that sequenced faba bean leaf transcriptome using the PacBio single-molecule long-read isoform sequencing platform could identify 28,569 nonredundant unigenes (Lyu et al., 2021) [50]. The predicted proteins and transcription factors included NB-ARC, Myb_domain, C3H, bHLH, and heat shock proteins, implying that this genome has an abundance of stress resistance genes [50]. In this study, we showed the complex regulation of the ABA and MAPK signaling pathways, amino acid synthesis, and carbohydrate metabolism in the drought stress response in the seed germination stage in faba bean.

Finally, we confirmed the role of the *AAT* gene in Asp metabolism in tobacco. The ectopic overexpression of DN8046_c0_g1 in tobacco significantly enhanced the tolerance to drought stress. The ectopic overexpression of *AAT* increased the activity of AAT and the contents of aspartate, proline, and arginine under normal conditions, confirming the success of gene transformation in tobacco. Furthermore, the activity of AAT and the contents of glutamate, cysteine, isoleucine, glycine, and arginine were further enhanced by drought stress. However, lower contents of glycine and arginine are beneficial to drought tolerance (Appendix A). Concurrently, the ectopic overexpression of *AAT* resulted in an increase in POD activity, Glutamate content, and relative water loss. Therefore, these results showed that the ectopic overexpression of faba bean *AAT* in *N. benthamiana* enhanced drought tolerance by increasing AAT activity and the regulation of amino acid metabolism, ROS scavenging, and relative water loss under drought stress. However, there should be further investigations into whether the *AAT* gene positively regulates drought tolerance in faba bean in the future.

## 4. Materials and Methods

### 4.1. Plant Materials and Drought Stress Treatment

The faba bean varieties E1 and C105 were provided by Institute of Food Crops, Hubei Academy of Agricultural Sciences. For NGS sequencing, the drought stress treatments were conducted according to the protocol described by Wei et al. (2023) [51]. In brief, seeds of E1 and C105 were surface-sterilized for 5 min in 10% hypochlorite, then washed 4 times with sterile water. Twenty-five treated seeds were placed in Petri dishes (12 cm diameter), with two layers of filter paper in each Petri dish, and soaked with 15 mL 10% concentration of mannitol solution with three replicates. Seeds treated with distilled water were used as controls. Petri dishes were regularly monitored at the level of containing solution and, if necessary, 10% concentration of mannitol solution or distilled H_2_O was added to retain a constant concentration. The Petri dishes were placed in a growth chamber under controlled conditions (25 °C, 16 h light/8 h dark). The 1/4 portion of cotyledons with embryos were excised from the germinating seeds at 16 h or 64 h, frozen in liquid nitrogen immediately, and kept at −80 °C for RNA-seq. For SMRT sequencing, seeds of E1 were treated with mannitol solution as mentioned above. The embryos of seeds were excised at 16 h and 64 h in a similar way, as described above, and the roots and stems with leaves of the seedlings at 6 d were excised separately. Then, they were immediately frozen in liquid nitrogen and kept at −80 °C for RNA extraction.

### 4.2. Library Construction for NGS

RNA from each sample was extracted individually (10 μg per tissue) using Trizol reagent (Invitrogen, Carlsbad, CA, USA) according to the manufacturer’s instructions. RNA purity was checked using the nanodrop 2000 spectrophotometer (LabTech, Boston, MA, USA). RNA integrity and concentration were assessed using the RNA Nano 6000 Assay Kit of the Bioanalyzer 2100 system (Agilent Technologies, Santa Clara, CA, USA). Twenty-four libraries were constructed, that is, the treated and control groups of E1 at 16 h and 64 h, each with three replicates (named as T1-16-1, T1-16-2, T1-16-3, T1-64-1, T1-64-2, T1-64-3, CK1-16-1, CK1-16-2, CK1-16-3, CK1-64-1, CK1-64-2, and CK1-64-3, respectively). The treated and control groups of C105 at 16 h and 64 h each had three replicates (named as T2-16-1, T2-16-2, T2-16-3, T2-64-1, T2-64-2, T2-64-3, CK2-16-1, CK2-16-2, CK2-16-3, CK2-64-1, CK2-64-2, and CK2-64-3, respectively).

### 4.3. Library Construction for Pacbio SMRT Sequencing

The embryos, roots, and stems with leaves of E1, which were collected from seeds at 16 h and 64 h under normal conditions and from seedlings at 64 h after drought stress treatment, respectively, were pooled to make one library. The RNAs were isolated as mentioned above, and equal amounts of RNA from each sample were mixed (5 μg per tissue) to construct the iso-seq libraries.

### 4.4. Illumina and Pacbio SMRT Sequencing

For Illumina sequencing, a total of 2 μg RNA per sample was used to construct libraries with NEBNext^®^ Ultra™ RNA Library Prep Kit for Illumina^®^ (#E7530L, NEB, Ipswich, MA, USA) following the manufacturer’s recommendations. Briefly, mRNA was purified from total RNA using poly T oligo-attached magnetic beads. Fragmentation was carried out using divalent cations under an elevated temperature in NEBNext first-strand synthesis reaction buffer (5×). First-strand cDNA was synthesized using random hexamer primer and RNase H. Second-strand cDNA synthesis was subsequently performed using buffer, dNTPs, DNA polymerase I, and RNase H. The library fragments were purified with QiaQuick PCR kits and eluted with EB buffer, then terminal repair, A-tailing, and adapter addition were implemented. The target products were retrieved, and PCR was performed, then the library was completed. The RNA concentration of the library was measured using the Qubit^®^ RNA Assay Kit in Qubit^®^ 3.0 to preliminarily quantify and then dilute to 1 ng/μL. The insert size was assessed using the Agilent Bioanalyzer 2100 system (Agilent Technologies, CA, USA), and the qualified insert size was accurately quantified using the StepOnePlus™ Real-Time PCR System (Library valid concentration > 10 nM). The clustering of the index-coded samples was performed on a cBot cluster generation system using the novaSeq PE Cluster Kit v4-cBot-HS (Illumina, San Diego, CA, USA) according to the manufacturer’s instructions. After cluster generation, the libraries were sequenced on an Illumina platform and 150 bp paired-end reads were generated.

For Pacbio SMRT sequencing, the Clontech SMARTER cDNA synthesis kit with Oligo-dT primers was used to generate first- and second-strand cDNA from polyA mRNA. Size fractionation and selection (>4 kb) were performed using the BluePippin™ Size Selection System (Sage Science, Beverly, MA, USA). The SMRT bell library was constructed with the Pacific Biosciences DNA Template Prep Kit 2.0 (NEB, Beverly, MA, USA), and SMRT sequencing was then performed using the Pacific Bioscience Sequel System.

### 4.5. Work Pipeline for Full-Length Transcript

The full-length transcript was obtained according to the work pipeline described as follows. Firstly, the adaptor contamination, low-quality bases, and undetermined bases in the readings produced by Illumine sequencing were removed using Fastp software v0.23.4 [52]. Secondly, FastQC was used to verify the quality of sequences (http://www.bioinformatics.babraham.ac.uk/projects/fastqc/) (accessed on 1 March 2023). The high-quality clean data with high Q20, Q30, and equitable GC content were used for downstream analyses. Thirdly, the raw data produced from SMRT sequencing were processed to generate the circular consensus sequence (CCS) reads using the software SMRTlink v13.1 [53,54]. Fourthly, the CCS reads were grouped as full-length or non-full-length reads based on the 5′ and 3′ end. The reads were considered as full-length if they had both the 5′and 3′ primers, as well as a poly(A) tail. Fifthly, transcript clusters for full-length reads were conducted by Iterative Clustering for Error Correction (ICE) through pairwise alignment and reiterative assignment. Finally, the high-quality isoforms were obtained after polishing the cluster consensus reads and correcting the nucleotide mismatches using the RNA-Seq data produced from 24 libraries using the software isoseq3 3.4.x. Soft BUSCO (version 3.1.0) was used to evaluate the completeness of the transcript data [55], and the longest transcript was considered as one candidate unigene, which was clustered to obtain the final unigenes with CD-Hit [56]. The prediction of the open reading frame (ORF) was conducted for each unigene by TransDecoder (Verdion 5.5.0) [57]. Unigenes were annotated by aligning against the Nr protein (http://www.ncbi.nlm.nih.gov/) and the SwissProt (http://www.expasy.org/resources/uniprotkb-swiss-prot) databases using Diamond and against the GO, KO, and KOG using eggNOG-mapper.

### 4.6. GO Enrichment Analysis

GO enrichment analysis provided all the GO terms that were significantly enriched in DEGs compared to the reference transcriptome background and filtered the DEGs that corresponded to biological functions. Firstly, all DEGs were mapped to GO terms in the Gene Ontology database (http://www.geneontology.org/), the gene numbers were calculated for every term, and significantly enriched GO terms in DEGs compared to the genome background were defined by a hypergeometric test. The calculated *p*-value was adjusted with FDR correction, setting FDR ≤ 0.05 as a threshold. GO terms meeting this condition were defined as significantly enriched GO terms in DEGs. This analysis was able to determine the main biological functions that DEGs perform.

### 4.7. Pathway Enrichment Analysis

Pathway-based analysis was conducted by blasting against the KEGG database (https://www.kegg.jp/kegg/kegg1.html) (accessed on 1 January 2000). The significantly enriched metabolic pathways or signal transduction pathways in DEGs were identified by comparing with the whole-genome background. The calculated *p*-value was obtained through FDR Correction, taking FDR ≤ 0.05 as a threshold. Pathways meeting this condition were defined as significantly enriched pathways.

### 4.8. PPI Network Analysis

The Search Tool for Retrieval of Interacting Genes/Proteins (STRING) online database (version 11.0; https://string-db.org/) (accessed on 8 January 2021) was applied to construct protein–protein interaction networks using Glycine max as the background.

### 4.9. qRT-PCR Analysis

The total RNA used for the RNA-seq was treated with RNase-free DNase (Invitrogen, Frederick, MD, USA) and reversely transcribed by the RevertAid™ First Strand cDNA Synthesis Kit (Thermo Fisher Scientific, Waltham, MA, USA). The qRT-PCR reactions were carried out with gene-specific primers (Appendix A) in the CFX96™ Real-Time PCR Detection System (Bio-Rad, Hercules, CA, USA) with a housekeeping gene, NADH dehydrogenase subunit 4 (NADHD4), as the internal control [4]. A 10 μL volume for each reaction contained 5 μL of SYBR Premix Ex Taq™ (TAKARA, Shiga, Japan), 2 μL cDNA template, 0.3 μL of each forward and reverse primer (10 μM), and 2.4 μL ddH_2_O. Triplicate reactions were conducted for each gene with the procedure: 95 °C for 30 s, 40 cycles of 95 °C for 5 s, and 57 °C for 30 s. The 2^−ΔΔCT^ method was adopted to calculate the relative gene expression level [10].

### 4.10. Ectopic Overexpression of AAT and RT-PCR Detection

The ectopic overexpression of AAT by PVX (*potato virus X*) was carried out as described by Han et al. (2021) [28]. The *AAT* ectopic overexpression were conducted with four-week old *N. benthamiana* plants in a growth room (24 °C, 16 h/8 h light/dark, 100 μmol·m^−2^ s^−1^ white light). Gene-specific primers were used to amplify the coding sequence region (Appendix A), which was inserted into the vector of PVX-LIC and confirmed by sequencing. The *AAT*-PVX construct was introduced into *Agrobacterium tumefaciens* GV3101, and then inoculated with *N. benthimiana* through the infiltration method. Five plants were inoculated for AAT-PVX and empty vector, respectively, and three biological replicates were set. The new developing upper leaves beyond the inoculated were collected for RNA extraction and RT-PCR analysis after 7 d of infiltration, and the plants were kept under drought stress by withholding water. After 11 d of withholding water, the plants were photographed for phenotype analysis. RT-PCR was conducted with gene- and internal-control-specific primers.

### 4.11. Physiological Parameter Measurements

The enzymatic activities of AAT, SOD, and POD, contents of MDA and amino acids, and relative water loss were determined with the new developing leaves after 7 d of withholding water. The enzymatic activities of SOD and POD, MDA content, and relative water loss were measured as described by Han et al. [28]. AAT activity was measured with the Glutamic-oxalacetic Transaminase (GOT/AST) Assay kit (Solarbio LIFE SCIENCE, Beijing, China). The amino acid content was detected with high-performance liquid chromatography (HPLC) at Mengxi bio-pharmaceutical Technology Limited Company (Suzhou, China). The samples were collected from three plants, and three biological replicates were performed for each physiological parameter.

### 4.12. Data Statistical Analysis

Data analyses were conducted using Microsoft Excel 2016 and SPSS 16.0 (IBM, Armonk, NY, USA). The one-way ANOVA test was used to detected the significance. The differences at the 0.05 significance level were determined with Tukey’s multiple comparison test.

## 5. Conclusions

In this study, a number of candidate DEGs were identified in response to drought stress in the germinating seeds of faba bean by combining the PacBio and Illumina RNA sequencing techniques. The DEGs were significantly enriched in six and nine KEGG pathways at 16 h and 64 h after drought stress, respectively, which were involved in the ABA and MAPK pathway, N-glycosylation, sulfur metabolism, sugar metabolism, etc. The ectopic overexpression of *AAT* in tobacco enhanced drought stress tolerance by increasing the activities of AAT and POD and the contents of cysteine and isoleucine and reducing the content of MDA and the water loss. Thus, this study provides a novel insight into the genetic response and candidate genes involved in the response to drought stress in faba bean.

## Figures and Tables

**Figure 1 ijms-25-08875-f001:**
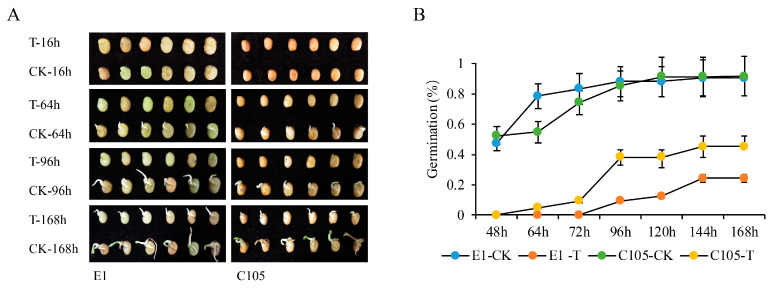
The phenotype and germination rate under normal and drought stress conditions. (**A**) The phenotype of faba bean E1 and C105 during germination. T-16h, CK-16h, T-64h, CK-64h, T-96h, CK-96h, T-168h, and CK-168h indicate that seeds were photographed at 16 h, 64 h, 96 h, and 168 h under drought stress or control conditions, respectively. (**B**) Statistics for the seed germinate rate. E1-T, E1-CK, C105-T, and C105-CK indicate that varieties E1 and C105 were treated under drought stress or control conditions, and 48 h to 168 h indicates the time of treatments.

**Figure 2 ijms-25-08875-f002:**
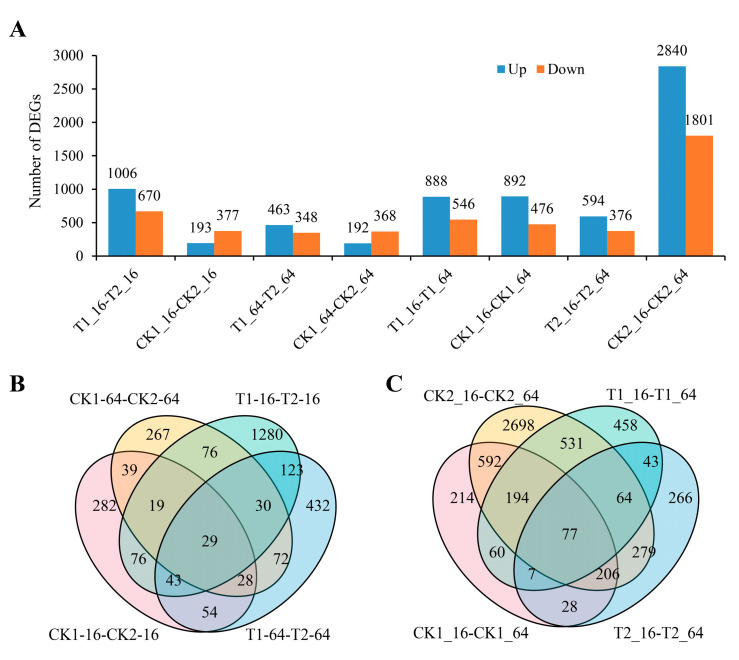
Quantitative and Venn analysis of the DEGs of two faba bean varieties under drought stress treatments. (**A**) Identified DEGs between different treatments. (**B**,**C**) Venn analysis of DEGs under different treatments. T1-16 and T1-64, E1 were treated for 16 h and 64 h, respectively; CK1-16 and CK1-64, E1 were under control conditions for 16 h and 64 h, respectively; T2-16 and T2-64, C105 were treated for 16 h and 64 h, respectively; CK2-16 and CK2-64, C105 were under control conditions for 16 h and 64 h, respectively.

**Figure 3 ijms-25-08875-f003:**
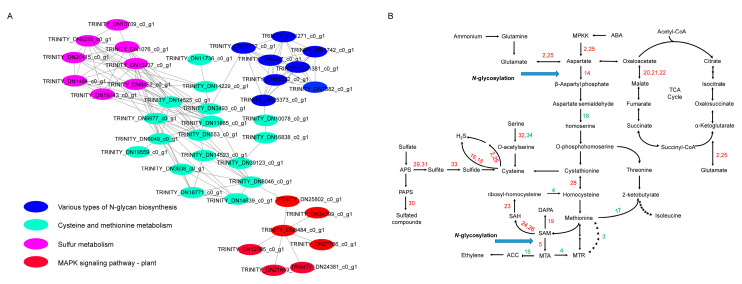
PPI network of DEG-encoding proteins in the comparison of T1_16 vs. T2_16 and their metabolic pathways. (**A**) PPI network. (**B**) The metabolic pathway of proteins presented in (**A**). The proteins represented by the numbers and the corresponding DEGs are as follows: 2, 25, Aspartate aminotransferase (DN8046_c0_g1, DN9977_c0_g1); 3, 1,2-dihydroxy-3-keto-5-methylthiopentene dioxygenase 1 (DN39123_c0_g1); 4, 5′-methylthioadenosine S-adenosylhomocysteine nucleosidase (DN16838_c0_g1); 5, spermidine spermine synthase (DN10078_c0_g1); 14, Aspartokinase (DN3493_c0_g1); 15, 1-aminocyclopropane-1-carboxylate deaminase (DN14229_c0_g1); 16, D-cysteine desulfhydrase 2 (DN11736_c0_g1); 17, Methionine gamma-lyase (DN11885_c0_g1); 18, aspartokinase homoserine dehydrogenase (DN553_c0_g1); 19, pyridoxal-phosphate-dependent aminotransferase (DN14523_c0_g1); 20, 22, Malate dehydrogenase (DN3638_c0_g1, DN14639_c0_g1); 21, L-lactate dehydrogenase A (DN16771_c0_g1); 23, Adenosylhomocysteinase (DN6049_c0_g1); 24, SAM-binding methyltransferase (DN19559_c0_g1); 26, Homocysteine s-methyltransferase (DN14525_c0_g1); 28, cysteine synthase cystathionine beta-synthase (DN13237_c0_g1); 29, 5′-adenylylsulfate reductase (DN1076_c0_g1); 30, 3(2),5-bisphosphate nucleotidase HAL2 (DN13709_c0_g1); 31, 5′-adenylylsulfate reductase (DN9235_c0_g1); 32, Serine acetyltransferase (DN20415_c0_g1); 33, Sulfite reductase ferredoxin (DN1464_c0_g1); and 34, Serine acetyltransferase (DN19743_c0_g1). ABA-signaling-pathway-related elements include ABA receptors (DN34399_c0_g1, DN27766_c0_g1, DN21659_c0_g1, and DN24381_c0_g1), phosphatase 2c (DN8484_c0_g1), and SNF1-like protein kinase 2 (DN12095_c0_g1). The MPKK is encoded by DN25802_c0_g1. N-glycosylation proteins include Dolichol-phosphate mannosyltransferase (DN25373_c0_g1), Dolichyl-diphosphooligosaccharide-protein glycosyltransferase (DN15532_c0_g1, DN7882_c0_g1, DN1381_c0_g1, and DN1271_c0_g1), GDP-Man:Man(3)GlcNAc(2)-PP-Dol alpha-1,2-mannosyltransferase (DN9307_c0_g1), Mannosyl-oligosaccharide 1,2-alpha-mannosidase MNS3 (DN1742_c0_g1), and UDP-N-acetylglucosamine--dolichyl-phosphate N-acetylglucosaminephospho transferase (DN11712_c0_g1). The red and green numbers indicate that the DEGs are up-regulated or down-regulated, respectively. Abbreviations: ACC, 1-aminocyclopropane-1-carboxylic acid; APS, 5′-adenylylsulfate; DAPA, 7,8-diaminopelargonic acid; MTA, 5′-methylthioadenosine; MTR, methylthioribose. PAPS, 3′-phosphoadenosine-5′-phosphosulfate; SAH, S-adenosylhomocysteine; SAM, S-adenosyl-l-methionine.

**Figure 4 ijms-25-08875-f004:**
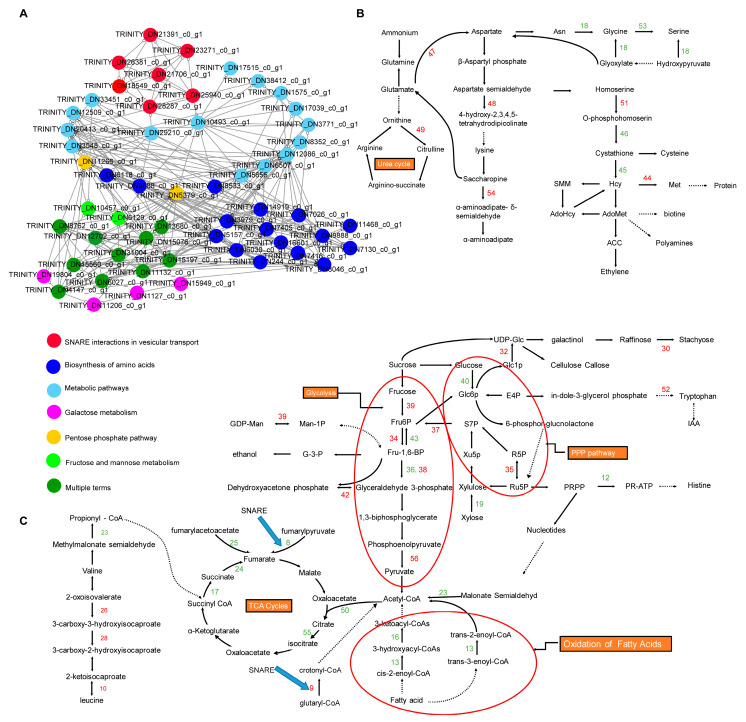
PPI network of DEG-encoding proteins in the comparison of T1_64 vs. T2_64 and their metabolism pathways. (**A**) PPI network. (**B**,**C**) The metabolic pathway of proteins presented in (**A**). The proteins represented by the numbers and the corresponding DEGs are as following: 8, Acylpyruvase FAHD1 (DN29210_c0_g1); 9, Glutaryl-CoA dehydrogenase (DN10493_c0_g1); 10, 2-oxoisovalerate dehydrogenase subunit alpha (DN8352_c0_g1); 12, ATP phosphoribosyltransferase (DN8533_c0_g1); 13, Enoyl-CoA hydratase isomerase (DN17039_c0_g1); 16, 3-hydroxyacyl-CoA dehydrogenase (DN1575_c0_g1); 17, Beta subunit of succinyl-CoA synthetase (DN6507_c0_g1); 18, Serine--glyoxylate transaminase (DN12086_c0_g1); 19, Xylose isomerase (DN6129_c0_g1); 20, aldehyde dehydrogenase (DN5655_c0_g1); 21, Glucose-6-phosphate 1-dehydrogenase-like protein (DN5379_c0_g1); 23, Methylmalonate-semialdehyde dehydrogenase (DN3548_c0_g1); 24, Succinate dehydrogenase iron-sulfur protein (DN20413_c0_g1); 25, Fumarylacetoacetase (DN12509_c0_g1); 26, 2-isopropylmalate synthase (DN9688_c0_g1); 27, Aldehyde dehydrogenase-like protein (DN11266_c0_g1); 28, Isopropylmalate isomerase (DN6118_c0_g1); 30, Raffinose synthase (DN1127_c0_g1); 31, Alpha-galactosidase (DN11206_c0_g1); 32, UTP-glucose-1-phosphate uridylyltransferase (DN19804_c0_g1); 33, Phosphoglucomutase (DN 4147_c0_g1); 34, 6-phosphofructokinase (DN6027_c0_g1); 35, Ribose-5-phosphate isomerase (DN13680_c0_g1); 36, Fructose-bisphosphate aldolase (DN11132_c0_g1); 37, Transaldolase (DN15076_c0_g1); 38, Fructose-bisphosphate aldolase (DN15560_c0_g1); 39, Mannose-1-phosphate guanyltransferase (DN10457_c0_g1); 40, Hexokinase (DN8762_c0_g1); 41, Fructokinase (DN31004_c0_g1); 42, Triosephosphate isomerase (DN15197_c0_g1); 43, Fructose-1-6-bisphosphatase (DN12702_c0_g1); 44, S-adenosylmethionine synthase (DN7416_c0_g1); 45, Cystathionine beta-synthase (DN11468_c0_g1); 46, Cystathionine gamma-synthase (DN3979_c0_g1); 47, Aspartate aminotransferase (DN8046_c0_g1); 48, Dihydrodipicolinate synthetase (DN16601_c0_g1); 49, Ornithine carbamoyltransferase Arg3 (DN14919_c0_g1); 50, Citrate synthase (DN6030_c0_g1); 51, THR1-homoserine kinase (DN9888_c0_g1); 52, Tryptophan synthase (DN7405_c0_g1); 53, Serine hydroxymethyltransferase (DN7026_c0_g1); 54, Saccharopine dehydrogenase (DN7130_c0_g1); 55, Aconitate hydratase 1 (DN244_c0_g1); and 56, Pyruvate kinase (DN5157_c0_g1). The red and green numbers indicate that the DEGs are up-regulated or down-regulated, respectively. Solid arrows refer to pathways with one reaction, and dashed arrows represent a multistep reaction.

**Figure 5 ijms-25-08875-f005:**
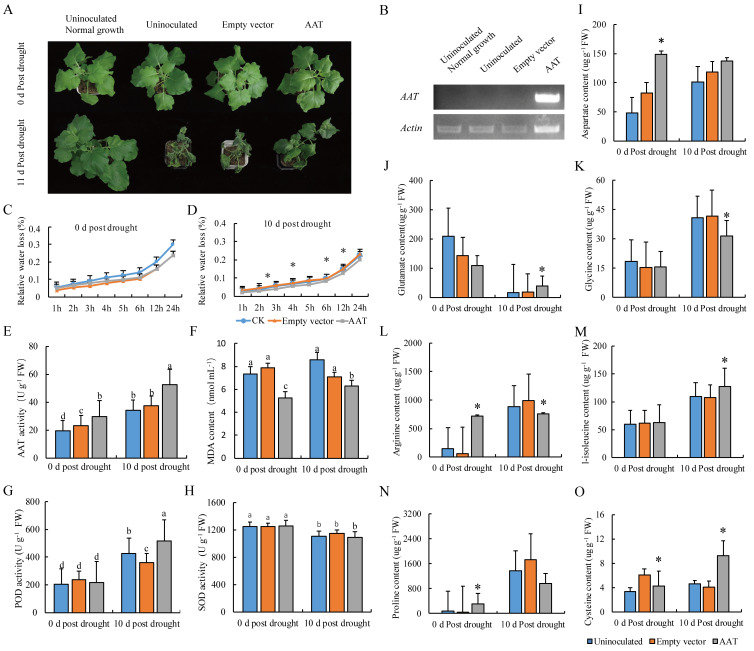
Phenotype and physiological analysis of tobacco plants with ectopic overexpressed *VfAAT*. (**A**) Phenotype of plants with overexpressed *VfAAT*, empty PVX vector, or uninoculated growth under normal conditions (0 d post drought) or drought stress conditions (11 d post drought). (**B**) The expression of *VfAAT* detected by RT-PCR. (**C**,**D**) Analysis of relative water loss. (**E**–**H**) Analysis of activity of AAT, SOD, POD, and content of MDA. (**I**–**O**) The content dection of aspartate, glutamate, glycine, arginine, 1-isoleucine, proline, and cysteine. FW, fresh weight; AAT, aspartate aminotransferase; MDA, malondialdehyde; SOD, superoxide dismutase; POD, peroxidase. The lowercase letters and * represent significant difference (*p* < 0.05) in the comparison of AAT with uninoculated and empty-vector plants.

**Figure 6 ijms-25-08875-f006:**
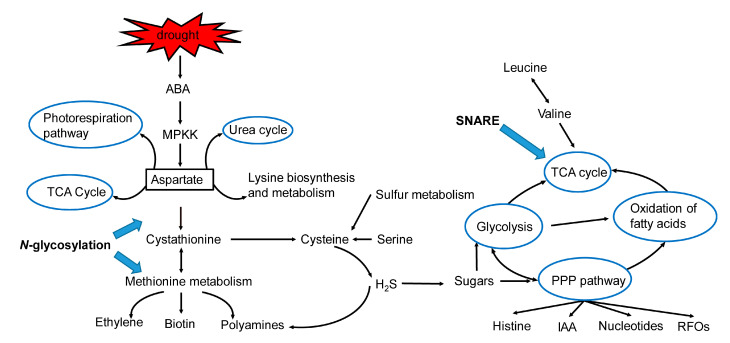
A schematic hypothetical model of key pathways and candidate genes under drought stress in the seed germination stage in faba bean. ABA, abscisic acid; IAA, indole-3-acetic acid; MPKK, mitogen-activated protein kinase; PPP, pentose phosphate pathway; RFOs, raffinose family oligosaccharides; SNARE, N-Ethylmaleimide-sensitive factor adaptor protein receptor; TCA, Tricarboxylic acid.

## Data Availability

The transcriptome sequencing data were submitted to the National Center for Biotechnology Information Sequence Read Archive [37] under Accession Number [DDBJ/ENA/GenBank] at National Center for Biotechnology Information (http://www.ncbi.nlm.nih.gov/) (GISP00000000).

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
