# Peer review of "Accurate Long-Read RNA Sequencing Analysis Reveals the Key Pathways and Candidate Genes under Drought Stress in the Seed Germination Stage in Faba Bean"

_ijms, 2024, doi:10.3390/ijms25168875_

Round 1
Reviewer 1 Report
Comments and Suggestions for Authors
The authors of the present manuscript have identified and compared the number of DEGs expressed in a sensitive to drought faba bean seeds variety E1 and tolerant ones C105 in response to drought stress in the germination stage at different time points using sequencing techniques.
KEGG pathways enrichment and PPI network analyses were performed. On the base of the obtained data, the authors have proposed a schematic model to explain the genetic regulation mechanism of the faba bean response to drought stress.
Additionally, the authors obtained that ectopic over-expression of faba bean gene AAT in N. benthamiana (using PVX as a gene vector) could improve the tolerance of this plant species to drought stress by increasing AAT activity, regulation of amino acids metabolism, ROS scavenging and the relative water loss under drought stress.
The present manuscript is interesting and gives a novel information about the expression of some genes included in regulation of drought tolerance of the faba bean to this stress.
The title exactly reflects the main idea of the study and the content of the article.
Abstract well summarized and presented the issues addressed.
Introduction presents the situation well, giving the definition of the issue to readers and its importance.
The methods are fully appropriate and are accurately explained.
The results and discussion well done.
The manuscript is well documented with many supplements.
The conclusions are well done.
I have some remarks.
English language needs a review in specific sections in the way the data are presented. Some sentences are excessively long and complicated.
1. On line 41 “faba bean” should not be in italic.
2. Review the sentence on lines 73-75 “…to a serious production reduction…”
3. The text from line 89 to 99 is not needed. “As a result, a total of…”. The same is given in the part “Conclusions”. I recommend omitting these sentences from the part “Introduction”.
4. The typo on line “C106” should be reviewed.
5. Please, check English of the legend under the Figure 1.
6. Please, insert the labels of the ordinate axis of Figure 2 (A).
7. A typo exists on line 407. The following sentence on lines 407-409 “carbohydrates serve as storage...” should start with a capital letter.
8. The sentence on lines 425-427 is not completed.
9. I recommend reviewing the typo on line 442 “regulate the of metabolisms”.
10. Please, review the text of the legend of Figure 6. “IAA” is duplicated.
Comments on the Quality of English LanguageEnglish language needs a review in specific sections in the way the data are presented. Some sentences are excessively long and complicated.
Author Response
The authors of the present manuscript have identified and compared the number of DEGs expressed in a sensitive to drought faba bean seeds variety E1 and tolerant ones C105 in response to drought stress in the germination stage at different time points using sequencing techniques.
KEGG pathways enrichment and PPI network analyses were performed. On the base of the obtained data, the authors have proposed a schematic model to explain the genetic regulation mechanism of the faba bean response to drought stress.
Additionally, the authors obtained that ectopic over-expression of faba bean gene AAT in N. benthamiana (using PVX as a gene vector) could improve the tolerance of this plant species to drought stress by increasing AAT activity, regulation of amino acids metabolism, ROS scavenging and the relative water loss under drought stress.
The present manuscript is interesting and gives a novel information about the expression of some genes included in regulation of drought tolerance of the faba bean to this stress.
The title exactly reflects the main idea of the study and the content of the article.
Abstract well summarized and presented the issues addressed.
Introduction presents the situation well, giving the definition of the issue to readers and its importance.
The methods are fully appropriate and are accurately explained.
The results and discussion well done.
The manuscript is well documented with many supplements.
The conclusions are well done.
I have some remarks.
English language needs a review in specific sections in the way the data are presented. Some sentences are excessively long and complicated.
Response: Thank you very much for your carefulness. We revised it this time.
- On line 41 “faba bean” should not be in italic.
- Review the sentence on lines 73-75 “…to a serious production reduction…”
- The text from line 89 to 99 is not needed. “As a result, a total of…”. The same is given in the part “Conclusions”. I recommend omitting these sentences from the part “Introduction”.
- The typo on line “C106” should be reviewed.
- Please, check English of the legend under the Figure 1.
- Please, insert the labels of the ordinate axis of Figure 2 (A).
- A typo exists on line 407. The following sentence on lines 407-409 “carbohydrates serve as storage...” should start with a capital letter.
- The sentence on lines 425-427 is not completed.
- I recommend reviewing the typo on line 442 “regulate the of metabolisms”.
- Please, review the text of the legend of Figure 6. “IAA” is duplicated.
Response:
- Thanks. We revised it.
-
Thanks. We revised it as: “Severe drought stress can cause non-uniform germination or non-germinating seeds after sowing, thus leading to a serious production loss or even no harvest”.
-
Thank you very much for your good idea. We revised it as:“ This study proves a novel insight into the key pathways and candidate genes for drought stress genetic improvement during seed germinations in this species”.
-
Thanks. We corrected it in Line 100.
-
Thanks. We had revised it in Fig.1 legend.
-
Thanks. We had revised the Figure 2 and added the axis of Figure 2A.
-
Thanks. We revised it in Line 406.
-
Thanks. We had revised it in line 424-426.
-
Thanks. We had revised it in Line 441.
-
Thanks. We had revised it in Line 453.
Comments on the Quality of English Language
English language needs a review in specific sections in the way the data are presented. Some sentences are excessively long and complicated.
Response: Thanks. We revised it.
Reviewer 2 Report
Comments and Suggestions for Authors
The manuscript presents a detailed and robust study on the identification of key pathways and candidate genes involved in the drought stress response at the seed germination stage of faba bean using advanced RNA sequencing techniques. The data generated provide valuable insights into the genetic mechanisms underpinning drought tolerance, which is highly relevant given the increasing prevalence of drought conditions due to climate change. However, several areas could benefit from further refinement and clarification.
Comments
The abstract is comprehensive but could be more concise. Consider focusing on the most significant findings and their implications. For example, the sentence "Identification of key genes in response to drought stress and the application to enhance the drought stress tolerance is efficient to cope with drought stress" can be simplified to improve readability.
The introduction provides a solid background but lacks a clear statement of the study's hypothesis or objectives. Adding a concise hypothesis will help readers understand the study's focus. References to previous studies on similar topics should be included to contextualize the current research within the existing body of knowledge.
There is a need for more detailed interpretation of the protein-protein interaction (PPI) network analysis results.
The study's findings on drought stress-responsive genes in faba bean could be linked to previous work on cold stress responses in the same species, as described in the manuscript "Identification of stress-related gene families in faba bean" (Scientific Reports, 2021)​. This would provide a broader understanding of stress responses in faba beans. For example, highlight any specific genes or pathways that have been previously identified in related studies, and discuss their potential overlap or differences with the drought-responsive genes identified in this study​.
Figure 3 and 4 need more detailed description in the main text. Ensure these figures are explicitly mentioned where relevant result and discussion.
Minor comment
The resolution and design of Fig. 2A need improvement. Figure appears pixelated and lack clarity
Comments on the Quality of English LanguageMinor editing of English language required
Author Response
The manuscript presents a detailed and robust study on the identification of key pathways and candidate genes involved in the drought stress response at the seed germination stage of faba bean using advanced RNA sequencing techniques. The data generated provide valuable insights into the genetic mechanisms underpinning drought tolerance, which is highly relevant given the increasing prevalence of drought conditions due to climate change. However, several areas could benefit from further refinement and clarification.
Response: Thank you very much for your critical comments and kind suggestions. We had revise our MS according to your suggestions.
Comments
The abstract is comprehensive but could be more concise. Consider focusing on the most significant findings and their implications. For example, the sentence "Identification of key genes in response to drought stress and the application to enhance the drought stress tolerance is efficient to cope with drought stress" can be simplified to improve readability.
Response: Thanks. We had revised it.
The introduction provides a solid background but lacks a clear statement of the study's hypothesis or objectives. Adding a concise hypothesis will help readers understand the study's focus. References to previous studies on similar topics should be included to contextualize the current research within the existing body of knowledge.
Response: Thank you very much. We had the objectives in the introduction in Lines 62-66 and 85-86 in the RED characters.
There is a need for more detailed interpretation of the protein-protein interaction (PPI) network analysis results.
Response: Thanks. We added the detailed explanation in the Lines 228-231
The study's findings on drought stress-responsive genes in faba bean could be linked to previous work on cold stress responses in the same species, as described in the manuscript "Identification of stress-related gene families in faba bean" (Scientific Reports, 2021)​. This would provide a broader understanding of stress responses in faba beans. For example, highlight any specific genes or pathways that have been previously identified in related studies, and discuss their potential overlap or differences with the drought-responsive genes identified in this study​.
Response: Thanks. We had added the reference and discussion in the Lines 443-450.
Figure 3 and 4 need more detailed description in the main text. Ensure these figures are explicitly mentioned where relevant result and discussion.
Response: Thanks. We had added it in the Lines 228-231 and 270-275.
Minor comment
The resolution and design of Fig. 2A need improvement. Figure appears pixelated and lack clarity
Response: Thanks. We had revised figure 2 and added the axis in the Figure 2A.
Comments on the Quality of English LanguageMinor editing of English language required
Response: Thanks. We revisede it.